# Development of Single-Cell Transcriptomics and Its Application in COVID-19

**DOI:** 10.3390/v14102271

**Published:** 2022-10-16

**Authors:** Chaochao Wang, Ting Huyan, Xiaojie Zhou, Xuanshuo Zhang, Suyang Duan, Shan Gao, Shanfeng Jiang, Qi Li

**Affiliations:** Key Laboratory for Space Biosciences & Biotechnology, Institute of Special Environmental Biophysics, School of Life Sciences, Northwestern Polytechnical University, Xi’an 710072, China

**Keywords:** single-cell transcriptomics, COVID-19, SARS-CoV-2, immune response

## Abstract

Over the last three years, the severe acute respiratory syndrome coronavirus 2 (SARS-CoV-2)-related health crisis has claimed over six million lives and caused USD 12 trillion losses to the global economy. SARS-CoV-2 continuously mutates and evolves with a high basic reproduction number (R0), resulting in a variety of clinical manifestations ranging from asymptomatic infection to acute respiratory distress syndrome (ARDS) and even death. To gain a better understanding of coronavirus disease 2019 (COVID-19), it is critical to investigate the components that cause various clinical manifestations. Single-cell sequencing has substantial advantages in terms of identifying differentially expressed genes among individual cells, which can provide a better understanding of the various physiological and pathological processes. This article reviewed the use of single-cell transcriptomics in COVID-19 research, examined the immune response disparities generated by SARS-CoV-2, and offered insights regarding how to improve COVID-19 diagnosis and treatment plans.

## 1. Introduction

In 2019, a new coronavirus known as severe acute respiratory syndrome coronavirus 2 (SARS-CoV-2) emerged, with incredible transmission speed and virulence, resulting in massive loss of life and monetary resources worldwide. SARS-CoV-2 has infected over 40 million people and caused more than 6 million deaths worldwide as of 20 March 2022. Controlling the spread of SARS-CoV-2 and reducing the number of deaths is a problem that all of humanity faces. Control measures such as quarantine, mask use, frequent hand washing, and avoidance of large gatherings are critical in preventing this pandemic, and understanding the pathogenic mechanism of COVID-19 and developing safe and effective vaccines will provide individuals with more protection [1]. Evidence has shown that there are different immune cell phenotypes between COVID-19 patients with severe conditions and those with mild symptoms [2]. Moreover, COVID-19 patients with severe expiratory dyspnea exhibit a complicated immune system dysfunction [3]. Therefore, elucidating the pathogenic mechanism of COVID-19 requires an understanding of the pathological profile of many tissues and cells as well as the differences in the immune response between asymptomatic and severely ill patients. Numerous studies have been conducted to investigate the pathological changes in peripheral blood and lung tissue cells of COVID-19 patients.

The concept of single-cell RNA sequencing (scRNA-seq) was first proposed in 2009 by Tang et al. [4]. Over the last decade, with advancements in cell isolation and nucleic acid amplification, sequencing techniques have become increasingly diversified [5]. Some of them are concerned with identifying more cells, while others are concerned with detecting more genes [6]. Sequencing technologies with different emphases can be chosen based on the features of the samples and the objective of the experiment. Single-cell transcriptomics, in comparison to conventional transcriptomics, can more clearly display the heterogeneity between cells [7] and, as a result, has been swiftly adopted by researchers.

At present, single-cell RNA sequencing is a critical tool for elucidating the pathogenic process of COVID-19. In the published literature, the peripheral blood cells and lung tissue cells of patients are frequently used samples by researchers. This article aims to review the applications of single-cell transcriptomics in COVID-19 to investigate the pathological alterations of tissue and cells among patients and to provide a new perspective on preventing and treating this pandemic.

## 2. Development of Single-Cell Transcriptomics

Since its inception in 2009, single-cell sequencing technology has emerged as a primary driver of sequencing technology development and has garnered most of the attention. However, early on, this approach could detect only a small number of cells, limiting its application. Two years later, Islam’s group pioneered the method of single-cell tagged reverse transcription (STRT) [8], which enabled high-throughput single-cell sequencing. Since then, single-cell sequencing technology has advanced significantly. Currently, single-cell research is mostly concerned with the transcriptome [9], spatial transcriptome [10], epigenetics [11] (DNA methylation [12], chromatin accessibility [13], chromatin interactions [14], histone modifications [15], and histone marks [16]), etc. The present article focuses mostly on single-cell transcriptomic research.

## 3. Upstream Data Acquisition

There are diverse single-cell-based sequencing technologies, but the fundamental stages are extremely similar. The first step is cell isolation, which is crucial to data quality. Then, RNA is extracted from each single cell. cDNA is obtained by reverse transcription. Finally, the amplified cDNA is applied to the sequencing platform [17] (Figure 1). Table 1 below summarizes the single-cell transcriptomic technologies and platforms.

Depending on the library-building procedures, the platform for single-cell sequencing can be separated into droplet-based and plate-based methods. Fluidigm(San Francisco, USA), 10X Genomics(California, USA), and Illumina(California, USA) are now well-known sequencing firms. In 2013, the first commercial platform for single-cell sequencing, Fluidigm C1, was introduced. Although it can extract the complete cDNA library for each cell, it can only simultaneously analyze 96 cells, which significantly restricts its application [29]. Using 10× Genomics Chromium Single Cell Gene Expression Solution, 10× Genomics built a cDNA library on the Illumina sequencing technology in 2016 to simultaneously identify tens of thousands of cells. The characteristics of Illumina’s two traditional sequencing devices, HiSeq and NextSeq, are detailed in Table 2. The advancement of single-cell sequencing technology hinges on the production of a full-length cDNA library and the identification of more cells. The two approaches, droplet-based 10× Genomics Chromium and a plate-based switching mechanism at the 5’ end of RNA template sequencing (SMART-seq), which are widely accepted and utilized by relevant practitioners, accurately illustrate these two points [30]. By using a water-in-oil microreaction system, the former recognizes distinct cells in a population through sequence tags and generates a digital gene map at the single-cell level, enabling the investigation of thousands of single-cell populations. This approach, however, has particular requirements regarding the number of samples and is not suitable for detecting low-biomass samples with few cells. The second generation of SMART-seq (SMART-seq2) is currently the most widely used plate-based technology for scRNA-seq. SMART-seq2 improves the original purification process by adding betaine to the reverse transcription (RT) step, which increases the thermostability of transcription enzymes (proteins) [18], decreases base pairing and the reliance on thermal melting through destabilizing DNA helices, and thus increases DNA yield. The use of Moloney murine leukemia (M-MLV) reverse transcriptase during the RT-PCR process allows for the binding of template-switching oligos (TSOs) and thus facilitates full-length sequencing [31]. However, the downside of this method is that it cannot distinguish forward and reverse strands, so the library construction efficiency is reduced when transcripts exceed 4 kb in length.

## 4. Downstream Data Analysis

Offline data formats vary between systems for sequencing. The offline data should be transformed to a particular data format, typically a gene expression matrix, with the row name as cell name, column name as gene name, and value as the expression level of the gene in the cell, once they have been collected. Following matrix quality verification, the subsequent analysis can commence. The vigorous advancement of sequencing technology has led to the rapid innovation of downstream bioinformatics analysis techniques, including dimensionality reduction techniques, cell annotation, and differential analysis. Dimension represents the number of gene types found in single-cell transcriptomics data. Dimensionality reduction techniques can be used to eliminate irrelevant data while retaining the essential information about cell differences, which is advantageous for cell clustering and visualization. The most frequently used strategies for reducing the dimensionality of single-cell data are t-distributed stochastic neighbor embedding (tSNE) and uniform manifold approximation and projection (UMAP) [32,33], which alternately focus on the local and global structure of the retained data. Additionally, the data that have been decreased in dimension can be clustered into cell clusters. The annotation of cell clusters is a vital step in exhibiting the biological meaning of data. SingleR is a widely used tool for cell annotation. By utilizing a reference transcriptome dataset of pure cell types, SingleR is able to infer the probable cell type of each individual cell autonomously, which significantly eliminates the subjectivity of manual annotation. Once the cell type has been established, the different cell types and the expression difference of the same gene can be compared between clusters. The differentially expressed genes are then frequently enriched in biological pathways to investigate the phenotypic and functional alterations by Gene Ontology (GO) analysis and Kyoto Encyclopedia of Genes and Genomes (KEGG) enrichment analysis. Additionally, depending on the goal of the experiment, extra analyses, such as pseudotime analysis, which can describe how gene expression changes over time [34], and RNA velocity analysis [35], which can describe the trajectory of cell differentiation, can be used.

## 5. Introduction to Downstream Analysis Tools

Most of the current analyses of single-cell transcriptomics are based on R and Python. With its powerful visualization and interaction capabilities, the R language is favored by most researchers. Therefore, this article will introduce the analysis tools based on the R language. The current mainstream R packages for single-cell transcriptomics study include Scater, Seuart, and Monocle (Table 3).

In 2017, Davis et al. realized that the existing data structure has difficulty in describing scRNA-seq data, so they proposed Scater, which facilitated data preprocessing, quality control, and visualization [36]. In Scater, quality control of cells and genes can be performed through the perCellQCMetrics and perFeatureQCMetrics functions. For example, cells with fewer expressed genes and genes expressed in very few cells can be filtered out by adjusting the specific threshold according to the research object. Subsequently, the filtered data are dimensionality reduction clustered through the runPCA and runTSNE functions. After clustering, the different categories of cells can be compared to obtain the desired information. The emergence of the Scater package greatly simplified tedious operations, but it still has some shortcomings: for example, Scater cannot remove the batch effect of the data independently, which needs to be performed with the assistance of the MNNs (mutual nearest neighbors) method in the scran package.

In 2015, Seuart was first proposed to achieve spatial mapping of 851 *Danio rerio* embryonic cells [37]. However, the version released in 2018 was the one that became widespread. In contrast with Scater, the third version of Seuart can remove the batch effect of samples independently by using the canonical correlation analysis (CCA) method, and its visualization function is more powerful [38]. Seuart uses a special data structure containing gene count data and metadata: SeuratObject. In SeuratObject, multiple data can be integrated through the FindIntegrationAnchors function and IntegrateData function. After passing the quality control of the PercentageFeatureSet function, the “PCA+tSNE” or “PCA+UMAP” mode can be used to perform dimensionality reduction. That is, PCA results are processed through tSNE or UMAP. In the fourth version, Seuart also provides a workflow named Azimuth to automatically annotate cell types, which provides a web application and is simple to operate for users with no programming experience [39]. After annotating cell types, the corresponding subset can be extracted by excavating the changes in a particular cell type through the subset function.

Different from the former two packages, Monocle can construct a developmental trajectory of cells [40]. Bifurcation of developmental trajectories is likely to be driven by pathological factors. Therefore, in developmental trajectories and nodes, the genes with changed expression in bifurcation events may be targets related to the disease. In Monocle, after creating the unique data structure CellDataSet, a set of specific genes can be selected as a basis for judging the state of cells. Then, the development trajectory can be constructed through the reducing Dimension and order Cells functions.

The aforementioned three packages can basically complete the whole single-cell transcriptome downstream analysis process alone, but in fact, other packages in some aspects can be used in the process to obtain better results (Figure 2). For example, Seuart can be used to integrate data, and then Scater and Monocle can be used to process them. The data processed by Scater and Seuart can also be transmitted to Monocle species to obtain the developmental trajectory. In addition, SingleR can be used in cell annotation.

Although these packages are functionally interchangeable, it is recommended that researchers select a package for the main body of the analysis and then replace it in individual steps rather than replacing most of them.

## 6. Application of Single-Cell Transcriptomics in COVID-19

Bulk RNA sequencing can identify the global changes in the tissue microenvironment caused by COVID-19. For instance, in a study employing bulk RNA sequencing on nasal swabs from COVID-19 patients, the authors discovered that children had higher levels of protective cytokines and stronger immune responses than adults. However, the origin of these cytokines remains unknown [41]. In another study focusing on the salivary glands of coronavirus patient, Chen et al. discovered that ACE2 was mostly expressed by gland cells [42]. However, bulk RNA sequencing hardly clarified whether the number of glandular cells altered or the expression of ACE2 in glandular cells increased during the course of COVID-19. Using bulk RNA sequencing, Thair et al. established a collection of gene signatures to assess COVID-19 and other viral disorders based on gene expression variations between patients and healthy controls [43]. However, bulk sequencing cannot identify the origin of genes with variable expression. It is plausible that the differentially expressed genes from majority cells are more significant than those from minority cells, and scRNA-seq can expose this information. In conclusion, single-cell RNA sequencing can reveal in which cell the expression of which genes has altered during the course of COVID-19. It will facilitate the design of drugs intended to target a certain gene in desired cells. Then the application of single-cell transcriptomics in COVID-19 were reviewed from the following five aspects.

## 7. Identification of Cell Types

Different immune cells respond differently to viral infection. For instance, several immune cells appear to be dysfunctional in patients with COVID-19 [44,45,46]. By analyzing the proportional changes in different subtypes within the same cell type, it is feasible to identify the cell type most associated with SARS-CoV-2 infection, thereby identifying potential targets for the prevention and treatment of COVID-19. As a result, one of the most common applications of scRNA-seq is to identify the most critical cell types during infection for subsequent research.

Zhu et al. clustered peripheral blood mononuclear cells (PBMCs) of COVID-19 patients into T cells, B cells, monocytes, natural killer (NK) cells, dendritic cells (DCs), stem cells, and megakaryocytes by using scRNA-seq. They discovered that in infected people, the proportion of circulating plasma cells, a type of B cell, was approximately five times greater than that in healthy volunteers, which may be explained by the fact that the body increased the number of plasma cells to fight the virus [47]. Through identifying the cell types, Lee et al. discovered that the degree of proportional change in immune cells was significantly greater in patients with severe COVID-19 symptoms than in those with mild symptoms [48]. The proportion of classical monocytes significantly increased, while the proportions of DCs, nonclassical monocytes, and NK cells significantly decreased (Figure 3). For example, the proportion of NK cells in severe COVID-19 patients was half of that in healthy donors. Compared to that in healthy donors, the proportion of NK cells in moderately ill patients was barely decreased. Notably, Aaron et al. detected a novel type of cell, developing neutrophils, in the PBMCs of SARS-CoV-2-infected persons with ARDS [49]. They assumed that these cells originated from plasmablasts but were highly heterogeneous in terms of gene expression and cellular complexity; therefore, they classified these cells as a new cell type. Regrettably, they did not further confirm this novel type of cell in biological experiments, but it is clear that this type of cell evolved during virus infection may play a significant role in COVID-19. Yan Zhang’s group also identified two new monocyte subtypes, Mono0 and Mono5, in the peripheral blood of severely/critically ill patients by scRNA-seq. These subsets expressed the amphiregulin (*AREG*), epiregulin (*EREG*), and cytokine interleukin-18 (*IL-18*) genes and appeared to exhibit profibrogenic and proinflammatory features [50], suggesting that they may be potential targets for COVID-19 treatment. Arunachalam’s group used a tool called phospho-CyTOF, which includes many markers of the cell surface and interior, to annotate cells. Using this tool, they initially identified 12 cell types and found that the frequency of plasmablasts in the PBMCs of infected people increased considerably. When they further annotated PBMCs to 25 cell categories, they found that the frequency of plasmacytoid dendritic cells (pDCs) was significantly reduced [51]. These findings imply that the precision of cellular annotations may affect the interpretation of the results of immunological changes produced by COVID-19.

In conclusion, the accurate identification of each cell type can shed light on the unique alterations of the immune response generated by SARS-CoV-2. The cells with significantly altered proportions or wholly emerging during COVID-19 may be potential therapeutic targets. However, there is still a lack of consistent criteria for evaluating the scientific validity of existing approaches for cell annotation. To aid in identifying new cell types, it is critical to establish quantitative criteria for determining what type of difference and how much difference qualification can be defined as a new cell type. In addition, once a novel cell type is discovered using sequencing, follow-up biological identification experiments are indispensable.

## 8. Detection of the Inflammatory Response

The inflammatory response is an important immune defense mechanism in the human body when encountering an infectious organism [52], and inflammatory factors refer to the cytokines involved in inflammatory reactions. Sufficient data have demonstrated that SARS-CoV-2 can induce cytokine storms in certain individuals, which has resulted in the loss of many lives worldwide. Interferons (IFN), interleukins (IL), chemokines, colony-stimulating factors, and tumor necrosis factor alpha (TNF-alpha) are considered to be the primary agents that contribute to cytokine storms (Figure 3) [53]. Therefore, examining alterations in their levels in peripheral blood may shed light on critical molecular pathogenic pathways underlying severe COVID-19. According to Xu et al., IL-6 levels were positively correlated with the neutrophil-to-lymphocyte ratio and negatively correlated with decreased CD^3+^, CD^4+^, and CD^8+^ T-cell counts [54], indicating that these cells may be the primary source of IL-6. Changfu Yao et al. found that in COVID-19 patients, in response to type-I IFN, immunocytes in the peripheral blood were activated to a hyperinflammatory immune state [55], which indicates that interferon plays an important role in the inflammatory response induced by SARS-CoV-2 infection.

These studies conclusively established that scRNA-seq is available for studying the variation in serum inflammatory factor levels prior to and during SARS-CoV-2 infection, which has great significance for elucidating the mechanism of COVID-19-induced cytokine storms. However, it is difficult to pinpoint which element or panel of factors is responsible for the severe symptoms associated with SARS-CoV-2 infection. If the correlation between cytokines and the inflammatory response can be precisely clarified, it may be important to reveal the mechanism of severe COVID-19 and develop new therapeutic strategies.

## 9. Enrichment Analysis of Differentially Expressed Genes

scRNA-seq is a high-throughput approach for determining gene expression changes at the cellular level. However, the abiotic noise inherent in single-cell data should be eliminated first to avoid masking truly differentially expressed genes. Currently available methods for removing abiotic noise are downsampling algorithms that can alleviate the influence of sequencing depth and batch effects, such as CCA, MNNs, and Scanorama. After validation in wet-laboratory biological experiments, one or more of the most differentially expressed genes can be further enriched in biological pathways, which can aid in understanding their participation in related molecular regulatory networks, as well as their biological activities and roles in diseases.

Xie et al. discovered that in CD8^+^ effector T cells derived from COVID-19 patients, the genes associated with the “IL-17 signaling pathway” and “response to toxic substance” were gradually downregulated, as were the genes enriched in “lymphocyte/T-cell activation” and “positive regulation of immune effector process” [56]. By studying the differentially expressed genes in peripheral blood mononuclear cells (PBMCs), Li’s team determined the 25 most representative genes to be used as gene signatures to predict the infection status in patients. These markers accurately predicted increased infectiousness among 13 infected people [57]. Huang’s study revealed that the upregulated genes in hospitalized COVID-19 patients were enriched in (1) viral genome replication and infection, (2) type-I interferon signaling (IFN-I), (3) mitogen-activated protein kinase (MAPK), (4) lymphoid–nonlymphoid cell interactions, and (5) major histocompatibility complex (MHC) class II protein complex [58]. It is important to note that the downregulation of MAPK pathway genes can indicate a patient’s recovery. Consequently, the development of drugs that target this pathway may be a potential method of treating COVID-19. In moderately ill individuals with COVID-19, the genes involved in the innate immune response, the defense response to virus, the response to type-I interferon, and the type-I interferon-related signaling pathway are increased in T cells compared to healthy controls. In addition, HLA class II genes (HLA-DRA, HLA-DRB1, and HLA-DRB5) were increased in T cells from critically or chronically sick individuals [59]. By evaluating 36 metabolic pathways and 12 metabolism-related signaling pathways, Qi et al. determined that the majority of metabolic pathways are activated in COVID-19 patients, whereas metabolism-related signaling pathways are inhibited. Several metabolic processes were changed in patients with mild and severe COVID-19 compared to healthy controls. This includes glucose metabolism, the tricarboxylic acid cycle (TCA cycle), and oxidative phosphorylation (OXPHOS) [60]. There may be a relationship between metabolic problems and the immune dysfunction observed in COVID-19 individuals. These research focused on the differential expression of genes in well-defined cell types. Zhang’s team investigated the differentially expressed genes in novel cell subtypes and discovered that profibrotic genes, including AREG, EREG, a disintegrin and metalloproteinase with thrombospondin motifs 2 (ADAMTS2), and IL-18, are strongly elevated in patients [50]. Hou’s group revealed sex-biased features in COVID-19 patients by gene set enrichment analysis (GSEA). The enrichment degree of differentially expressed genes in 14 pathways, such as Toll-like receptor (TLR) signaling pathways, the RIG-I-like receptor signaling pathway, cytokines and growth factors, and the IL-17 signaling pathway, was significantly higher in the monocytes of male patients than in those of female patients [61]. This discovery lay the framework for understanding the immune responses underlying sex differences and devising sex-specific patient care strategies. Li et al. discovered, through the study of recovered patients of varying clinical severity, that the processing and presentation of antigen and intestinal immune network-related IgA production were downregulated in CD14^+^ monocytes of severely/critically ill recovered patients compared to healthy controls. In addition, HLA-DRA and HLA-DRB1 were increased in CD14^+^ monocytes and the dendritic cells of patients who had recovered from mild/moderate illness [62]. These data highlighted the immunological characteristics of peripheral blood cells during the patients’ convalescence, contributing to an improved prognosis for COVID-19. Compared to normal controls, the T-cell activation response in COVID-19 patients was substantially increased, whereas B-cell activation and differentiation were unexpectedly decreased [63]. These results suggested that cellular immunity may be more important than humoral immunity in SARS-CoV-2 infections.

The preceding studies explored the differentially expressed genes expressed by immune cells in the peripheral blood of COVID-19 patients and provided a biological foundation for revealing the immune response in COVID-19 patients. However, most of these studies focus more on the changes in specific pathways. Subsequent studies on the integrity of multiple pathways, such as those that can be employed as markers for COVID-19 prediction, are needed.

## 10. Recognition of Pathogen–Cell Interactions

Pathogenic microorganisms can disrupt the biological processes of host cells; for example, during the COVID-19 pandemic, patients may be coinfected with other viruses or bacteria. In this case, immunological dysfunction is the result of numerous viruses acting in concert. Thus, elucidating the mechanism of virus–cell interactions may reveal new avenues for patient recovery, and how to study the interaction between cells and microbes has received the most attention.

Based on the STAR aligner [64], Bost et al. proposed a new computational tool called Viral-Track, which can project scRNA-seq data to a high-quality database [65]. By using this approach, they found that there was infection of human metapneumovirus (hMPV), a virus that can infect the respiratory tract, in the bronchoalveolar lavage of COVID-19 patients. Then, they studied the immune landscape after the interaction of SARS-CoV-2 and hMPV, which showed that immune activation was suppressed in the monocyte compartment [66]. Wei Zhang’s team developed an algorithm called PathogenTrack that is based on unsupervised identification of features of the intracellular microbiota extracted from scRNA-seq data. Using this algorithm, they found that a small percentage of cells in the bronchoalveolar lavage fluid (BALF) samples of a COVID-19 patient, including neutrophils and macrophages, were infected with *Haemophilus parahemolyticus* [67], a bacterium that is usually associated with ARDS and septic shock [68].

## 11. Identification of Novel Biomarkers

Biomarkers are biochemical indicators that can reflect structural or functional changes at the organ, tissue, cell, and subcellular levels and are widely employed in biological and medical research [69]. With the rapid advancement of scRNA-seq technology, obscure biochemical signs have become readily visible at single-cell resolution. The more markers of COVID-19 that are discovered, the more clearly the pathogenic mechanism may be elucidated.

Wu et al. found that the reduction in IFN-I and the increase in PAI-I levels in the lungs of COVID-19 patients were associated with disease severity, implying that IFN-I and PAI-1 expression might be employed as predictors of COVID-19 severity [70]. By comparing the quantitative differences in the cell types in the lungs of patients to different degrees, Zhao et al. found that TM4SF1^+^ and KRT5^+^ lung progenitor cells were significantly expanded in the BALF of critically ill COVID-19 patients compared to those in healthy controls [71]. Another study revealed sex-biased transcriptional activation in SARS-CoV-2-infected macrophages in BALF, while toll-like receptor 7 (TLR7) and bruton tyrosine kinase (BTK) expression in monocytes may serve as biomarkers for predicting poor prognosis in male patients [61].

Through the application of scRNA-seq in the five aforementioned elements, two viewpoints on the features of COVID-19 can be shown. In terms of cell variation, the proportion of circulating plasma cells and classical monocytes increased dramatically in COVID-19 patients, but the proportion of DCs, nonclassical monocytes, NK cells, and certain lung progenitor cells fell significantly. From the perspective of gene variation, AREG, EREG, and HLA class II genes were considerably elevated in patients. More gene modifications are characterized by pathways. In patients, genes associated with the “IL-17 signaling pathway”, “response to toxic substance”, “lymphocyte /T-cell activation”, and “positive regulation of immune effector process” are downregulated, whereas genes associated with “viral genome replication and infection”, the “IFN-I signaling pathway”, and “MAPK” are upregulated.

Overall, a significant number of rigorous “wet experiments” are required to establish the role of these cells or genes in COVID-19 in order to discover the prospective therapeutic targets for COVID-19. 

## 12. Discussion

Since its inception, single-cell technology has significantly aided the advancement of the biomedical field. In the past, bulk RNA sequencing has revealed the entire expression difference in almost all cell types. Single-cell sequencing allows individual cells to be annotated, which means the changes in specific cell types under pathological conditions can be explored. The interpretation of SARS-CoV-2-induced immune dysfunction through the single-cell transcriptome has greatly improved our understanding of the prevention, precision therapy, and prognosis prediction of COVID-19.

However, single-cell sequencing technology has a long way to go in many areas. First, the transcription status of sequenced cells is inconsistent, and the immediate results cannot accurately represent the whole transcription map [72,73]. Second, gene deletions during nucleic acid amplification frequently complicate downstream analysis: it is unknown if the gene was not amplified due to technical error or was not expressed in the first place. However, there are currently some algorithms, such as Markov affinity-based graph imputation of cells (MAGIC) [74] and single-cell analysis via expression recovery (SAVER) [75], that can mitigate this disadvantage. If the efficiency of acquiring nucleic acid transcription from upstream data can be greatly improved, the quality of single-cell data will also be greatly improved. The majority of scRNA-seq technologies begin at the 3’ or 5’ end of the transcript and sequence to a depth of approximately 1 million reads [25]. There is still much room for improvement in terms of sequencing depth. Additionally, batch effects are inevitable when integrating and analyzing the data obtained from different platforms. Eliminating the technical noise while retaining actual biological properties is the next research direction for single-cell sequencing technology in the future. In addition, further reducing the cost of sequencing may make it more acceptable to researchers.

On the basis of single-cell transcriptome data, various novel views have been applied to assess the immunological profile of SARS-CoV-2 infection. For instance, identifying the most prevalent alterations among the expression of genes or biological pathways in patients with COVID-19 may provide direction for developing COVID-19 preventative and therapeutic techniques. Machine learning, one of the most prominent directions today, can extract internal logic from massive amounts of data to make accurate predictions. As a result, machine learning may be a viable option for reanalyzing the extracted transcriptome characteristics and predicting COVID-19 prognosis. Finally, it may be more informative to track the transcriptome alterations in a single patient over time rather than comparing the transcriptomes of healthy and infected individuals.

## Figures and Tables

**Figure 1 viruses-14-02271-f001:**
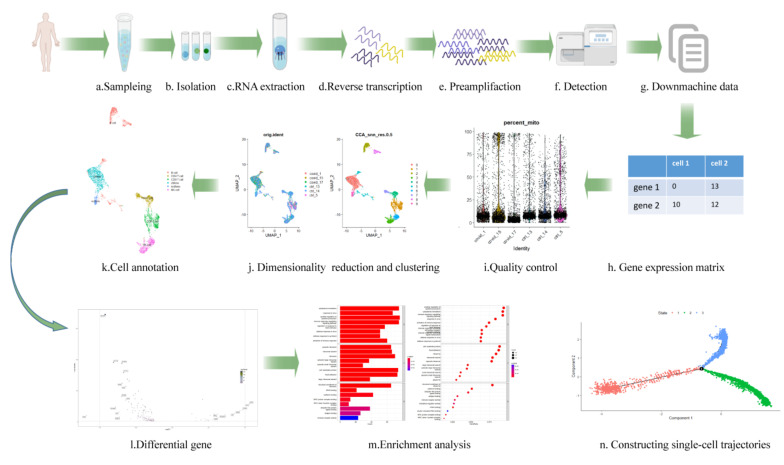
The acquisition and analysis of single cell data. The first row depicts the entire data collecting process (**a**–**g**). The second and third lines represent the procedure of downstream analysis. After build the gene expression matrix (**h**), quality control is carried out according to the proportion of mitochondrial genes (each point represents one cell) (**i**). The cells whose mitochondrial genes account for more than 10% are generally filtered. Subsequently, dimension reduction and clustering are carried out. The single cell data are projected onto two-dimensional plane (each point represents a cell). The dots with different colors on the left represent different samples, and the dots on the right represent different cell clusters after algorithm clustering (**j**). Then, the Singler package is used to annotate the cell clusters, and each color represents one cell type (**k**). The volcano map is a widely accepted method to visualize the differential expressed genes. Each dot represents one gene, and the dotted lines represent thresholds set artificially (**l**). The enrichment of the differential genes into the pathway is represented in (**m**). The vertical axis represents the name of the pathway, while the horizontal axis represents the number of genes in the pathway, and the color represents significance (**m**). Finally, the trajectory can be inferred according to the arrangement of the dots, and the bifurcation indicates that the trajectory of cell development has changed (**n**). This image was made by FigDraw (www.figdraw.com, (accessed on 27 August 2022)).

**Figure 2 viruses-14-02271-f002:**
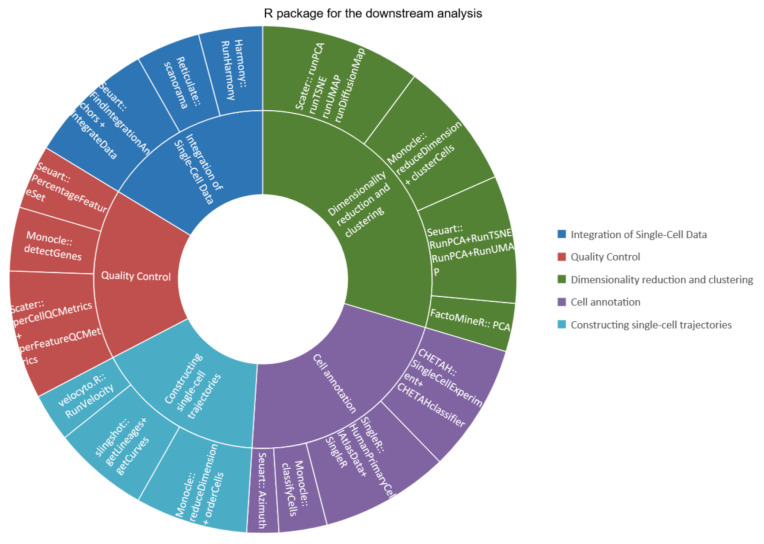
R package for the downstream analysis. The inner circle represents the main links of downstream analysis, and the outer circle represents the packages and functions that can realize these links.

**Figure 3 viruses-14-02271-f003:**
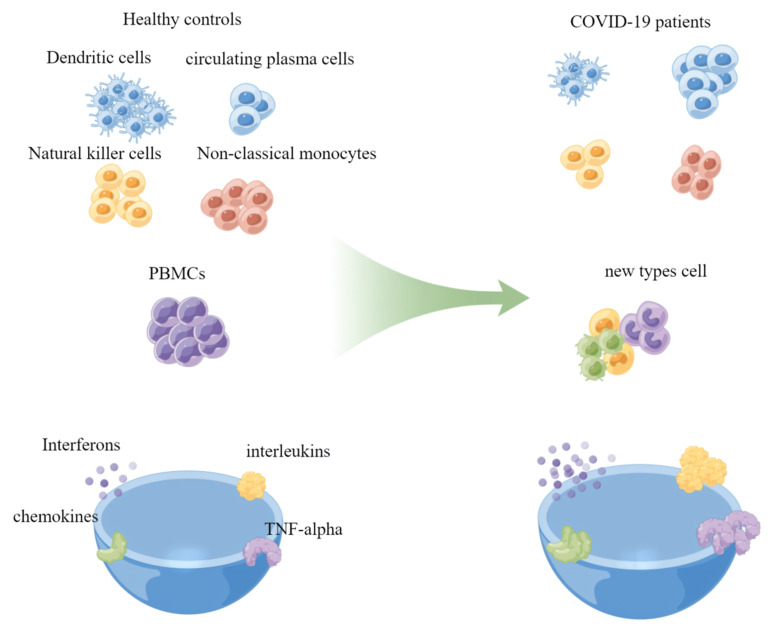
Cellular and inflammatory factor changes. The cell count simply reflects the increase or decrease in the cell proportions, as with the inflammatory factors. This image was made by FigDraw (www.figdraw.com, (accessed on 27 August 2022)).

**Table 1 viruses-14-02271-t001:** Technologies and platforms for single-cell transcriptomic analysis.

Technology	Platform	Sensitivity	Coverage	Throughput	References
SMART-Seq 2	Plate-based, Fluidigm C1 Illumina HiSeq 2000	Extremely High	Full length	Low	[18,19]
MARS-seq	Plate-based	Low	3′-end	High	[20,21]
Drop-seq	10X Genomics, Illumina NextSeq	Low	3′-end	High	[22,23]
CEL-seq 2	Fluidigm C1, Illumina TrueSeq	High	3′-end	Low	[24,25]
Seq-well	10X Genomics, Illumina NextSeq	Low	3′-end	High	[26,27]
SPLit-seq	Plate-based, Illumina NextSeq	Low	3′-end	Extremely High	[28]

**Table 2 viruses-14-02271-t002:** The features of HiSeq and NextSeq.

	HiSeq2000	NextSeq1000 and 2000
Maximum Read Length	2 × 100 bp	2 × 150 bp
Maximum Output	600 GB	360 GB
Runtime	~11 days	24–48 h
Reads	6 Billion (Paired-end Reads) 3 Billion (Single Reads)	2.4 Billion (Paired-end Reads) 1.2 Billion (Single Reads)
Quality Scores	>85% (2 × 50 bp) >80% (2 × 100 bp)	>=90% (2 × 50 bp) >=85% (2 × 150 pb)

**Table 3 viruses-14-02271-t003:** The functions of the three mainstream r packages.

	Scater	Seuart	Monocle
Integration of Single-Cell Data	×	√	×
Quality Control	√	√	√
Dimensionality reduction and clustering	√	√	√
Cell annotation	×	√	√
Differential expression analysis	√	√	√
Constructing single-cell trajectories	×	×	√

Note: ×: unavailable, √: available

## Data Availability

Not applicable.

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
