# Peer review of "Development of Single-Cell Transcriptomics and Its Application in COVID-19"

_viruses, 2022, doi:10.3390/v14102271_

Round 1
Reviewer 1 Report
The article was written nicely and describe the role of single cell RNAseq in identifying different cell types and differential expression of genes in different cell types during SARS-Cov2-infection. It would be helpful to understand in a better way if the authors able to provide any way to integrate all those data from single cell RNAseq and how we can identify the master gene/ or cell type for therapeutic target or for clinical application. Because so many genes are differentially expressed during infection, therefore it is hard to target all those genes simultaneously. Identification of the master molecule will help us to understand the mechanism of disease progression.
Author Response
Thanks for your valuable comments. The integration of transcriptome data should be considered from two aspects. One is the integration between scRNA data, it can be achieved by using Seurat package which we have mentioned in text (second paragraph on page 7). With regard to integrate scRNA data and bulk RNA data, to our knowledge, there are few methods to directly integrate the two types of data till now. Theoretically, supposing that adequate bulk RNA sequencing data of specific cell can be obtained, the algorithmic specialists can determine the gene expression mode of this cell to develop more accurate software for the annotation of scRNA data. We expect that more useful tools to integrate different sequencing data can be developed, which will greatly boost the development of bioinformatics research. Finally, thank you again for your valuable comments.
Reviewer 2 Report
Wang et al. described the process of single-cell RNA sequencing and the different systems in use to date. In addition, given the complex pathogenetic mechanism of COVID-19, they focused on the use of this approach in SARS-CoV-2 infection.
The manuscript covers topics that are current and will become increasingly important in the future. Despite this, some points need to be clarified:
- 3. Upstream data acquisition:The table referred to in this paragraph shows different sequencing systems, but then only a few platforms are discussed in the text. In order to better understand the different systems, it would also be good to report schematically the advantages and disadvantages of all of them.
- 4. Downstream data analysis: In paragraph thus described is not very clear and is, especially in the initial part, disconnected from the previous paragraph. Since the focus is on single-cell sequencing perhaps it would be better to start there and then discuss dimensionality and the relationship between the two. Is dimensionality a limitation of single-cell sequencing?
- Figure 1: The first part figure is very clear and useful to understand well the process from sample to sequencing. However, the second part should be described better by also a new figure that schematically describes the analysis steps required to arrive at the final data starting from the instrument output.
- The paragraphs regarding COVID-19 contain so much information, for a better and clear understanding of them it would be good to make a summary table that reported: scope of interest (example cell type/inflammatory response), application of sgRNAseq (cluster PBMCs), final results.
